# The Effectiveness of Equine Therapy Intervention on Activities of Daily Living, Quality of Life, Mood, Balance and Gait in Individuals with Parkinson’s Disease

**DOI:** 10.3390/healthcare10030561

**Published:** 2022-03-17

**Authors:** Anna Berardi, Giorgia Di Napoli, Monica Ernesto, Giovanni Fabbrini, Antonella Conte, Gina Ferrazzano, Fabio Viselli, Giovanni Galeoto

**Affiliations:** 1Department of Human Neurosciences, Sapienza University of Rome, Viale dell’Università, 30, 00185 Rome, Italy; anna.berardi@uniroma1.it (A.B.); giovanni.fabbrini@uniroma1.it (G.F.); antonella.conte@uniroma1.it (A.C.); gina.ferrazzano@uniroma1.it (G.F.); 2Sapienza University of Rome, Piazzale Aldo Moro, 5, 00185 Rome, Italy; dinapoli.1699457@studenti.uniroma1.it; 3Hospital San Govanni Battista, 30, 00185 Rome, Italy; m.ernesto@acismom.it (M.E.); f.viselli@acismom.it (F.V.); 4IRCSS Neuromed, Via Atinense, 18, 86077 Pozzilli, Italy

**Keywords:** rehabilitation, Parkinson’s disease, disability, animal-assisted therapy

## Abstract

Objectives: The objective of this study was to evaluate the efficacy of equine therapy (ET) to detect changes in the activities of daily living, quality of life, mood, balance, and gait in individuals with Parkinson’s disease (PD). Material and Methods: In the study, 17 participants with PD were recruited to participate in 10 sessions of ET. The inclusion criteria of the study were: second and third stages of the Hoehn and Yahr scale, Mini-Mental State Examination (MMSE) greater than or equal to 24 points, and age up to 85 years. The outcome measures administered at the beginning and the end of treatment relied on measurements from the Rivermead ADL scale, Parkinson’s Disease Questionnaire-39 (PDQ-39), Zung Self-Rating Depression Scale (SDS), unified Parkinson’s disease rating scale (UPDRS), and Tinetti balance assessment. Data from the stabilometric platform were also collected to objectify the value obtained by the Tinetti balance assessment. The ET program included 10 biweekly 45 min sessions. Results: The results obtained included statistically significant increases in measurements from the Rivermead ADL, PDQ-39, UPDRS, SDS, and Tinetti balance assessment scales. The stabilometric platform did not report significant changes in data. Conclusion: ET that was used as a supportive therapy for traditional treatments resulted in statistically significant improvements in the occupational performance, mood, quality of life, gait, and balance of the participants. Data from the stabilometric platform did not show significant changes.

## 1. Introduction

Parkinson’s disease (PD) is a progressive and degenerative neurological disease that is characterized by tremor, retropulsion (a tendency to fall backwards), rigidity, postural instability, bradykinesia, and hypomimia or amimia. These are often accompanied by other motor symptoms, such as voice disturbances and swallowing and walking impediments, as well as non-motor symptoms, such as mood disturbances and cognitive impairments, which may begin years before the other symptoms [1]. The presence of motor and non-motor symptoms and functional and social limitations that cause deterioration in the quality of life of individuals and caregivers makes a timely intervention, not only with pharmacological treatment, but also with rehabilitation therapies, even more important [2].

Recently, innovative rehabilitative techniques such as virtual reality, robot-assisted therapy, and other unconventional therapies have become more common [3,4]. Animal-assisted therapy (AAT) is a therapeutic intervention aimed at the treatment of disorders of the physical sphere, including neurological, psychomotor, cognitive, emotional and relational disorders. The therapy is meant for individuals with physical, psychological, sensory or neurological pathologies. The intervention is individualized, it must be medically indicated, and it is based upon a team approach that considers the person and his or her specific treatment goals and objectives [5,6]. Animals used in the interventions include dogs, cats, horses, rabbits, and donkeys. These are animals that can provide an empathic and emotional response in their interactions with humans [5]. Scientific investigations on the human-animal bond and its potential for health interventions have increased in the last decades; this suggests that these therapies can provide a wide range of physiological and psychosocial benefits [7,8]. Among the other types of therapy, equine therapy (ET) has shown benefits in children with autism spectrum disorder (ASD), infants with infantile cerebral paralysis, and people with multiple sclerosis. Children with ASD showed improvements in postural stability, receptive communication, coping, and participation in daily activities. People with multiple sclerosis showed a significant improvement in balance, fatigue, spasticity, and quality of life [9]. Finally, children with cerebral palsy showed improvements from ET, including gross motor function, functional performance, postural control, and balance [10,11].

In the present study, it was decided to activate an ET project with people suffering from PD because horses offer physical and psychological therapy, both of which allow for physical, tactile, and emotional stimulation [12]. All these characteristics offered participants with PD a treatment aimed at improving balance and posture, gait, rigidity, and involuntary movements, as well as making it easier to navigate social-relational and behavioral spheres, act with self-control and pay attention. Self-awareness, self-esteem, motivation, and volition were all enhanced. Also, the context in which the treatment was performed played an important role because it took place in an external environment, not a hospital, and the well-qualified people who assisted participants did not wear hospital uniforms. All these elements helped to encourage people to become active partners in their own therapeutic interventions.

## 2. Methods

This outcome research study was conducted by healthcare professionals at Sapienza University of Rome and ROMA—Rehabilitation & Outcome Measures Assessment Association. The research group has carried out many outcome measures in Italy [13,14,15,16,17,18,19,20,21,22,23].

### 2.1. Participants

The present study included participants diagnosed with PD who were recruited at the San Giovanni Battista Hospital in Rome, Italy, between February 2017 and February 2018. Before starting the study, the participants were asked to view and sign a form for informed consent, for acceptance and acknowledgment of the specific treatment of assisted intervention with the animals, for the use of personal data, and for the creation of video and photographic materials [24,25]. The inclusion criteria considered were: age up to 85 years, second and third stages of the Hoehn and Yahr scale, and Mini–Mental State Examination (MMSE) greater than or equal to 24 [26,27]. People older than 85, with an MMSE value lower than 24 and with H&Y stage equal to I or VI, were excluded from this study, moreover, attendance was required at least at 85%.

### 2.2. Assessment Tools

In the present study, primary outcomes were defined as the improvement of quality of life, performance in activities of daily living (ADL), and the tone of mood. Secondary outcomes included changes in balance and gait. The tests administered in this study included five tools: Parkinson’s Disease Questionnaire-39 (PDQ-39) [28]; Rivermead ADL scale [29]; Zung Self-Rating Depression Scale (SDS) [30]; Tinetti balance assessment [31] and unified Parkinson’s disease rating scale (UPDRS) [32].

The PDQ-39 was used to evaluate the change in the quality of life before and after the intervention. PDQ-39 consists of 39 items, and it is divided into eight subscales: mobility, ADL, emotional well-being, stigma, social support, cognitive faculties, communication and physical discomfort [28,33].

The Rivermead ADL scale was used to evaluate changes in participants’ skills in self-care at the beginning and at the end of the intervention. It was divided into three sub-scales: 16 for self-care, 9 for Level I household activities, and 6 for Level II household activities. In this study, only the score for items related to self-care was considered [29,34].

The SDS was used to assess mood changes, depression, and anxiety before and after the study [30,35].

The Tinetti balance assessment tool was used to evaluate changes in balance and gait before and after surgery in an environment with disconnected and uneven terrain. It was divided into two sub-scales: 9 to evaluate the balance and 7 to examine the gait. The result corresponds to the sum of the scores obtained by the two sub-scales and is closely related to the risk of falling [31,36].

The UPDRS, used to assess the severity of symptoms and the patient’s prognosis, was administered by the reference neurologist [37,38]. In the present study, only the score concerning the evaluation of daily life activities was included.

In addition to the assessment scales, the stabilometric lux postural platform (global postural system, Udine, Italy) was used to compare and objectify the score obtained with the Tinetti balance assessment tool. It consists of three load cells arranged on a circumference at an angular distance of 120°, and it obtains indications on the position of the patient’s center of gravity before proceeding with an analysis of stability [39]. The variables measured were the following: Middle coronal center (DX-SX) (unit); Middle sagittal center (PST-ANT) (unit); Curved length (unit); 90% confidence ellipse area (unit); Average speed (unit/ms); Sampling frequency (Hz); Fundamental harmonic frequency (DX-SX) (Hz); Fundamental harmonic frequency (PST-ANT) (Hz).

### 2.3. Intervention

The ET program included ten 45-min-long sessions that took place twice a week. Before starting full activities with the horse, they were required to carry out activities of daily life and to report on how things went; such activities included wearing single-use overalls, changing shoes, and sanitizing hands. The next step was organizing the setting according to the characteristics of each participant, such as physical ability. The new setting was introduced at the first session, and this was followed by the approach and knowledge of the horse that, through the therapist’s mediation, allowed for the establishment of a relationship based on trust and mutual understanding between participant and horse. The activities included:A caring and cleaning activity that allowed the participant to develop motor skills and coordination, cognitive skills (attention, memory), proprioception, and social–relational skills (Figure 1A).A horse-assisted walking activity that could be carried out in different ways, including inside or outside the equestrian area, and on a path, with or without obstacles, on which the individual was asked to define times of pause and resumption of activity (Figure 1B).A feeding activity that allowed the participant to develop relational and behavioral skills, with the creation of an even greater understanding with the animal (Figure 1C).Vocalisation exercises to call the horse or provide commands during the activities, which contributed to increasing self-confidence.Breathing exercises, assisted by the horse, with the aim of relaxing and reassuring the participant (Figure 1D).

Other activities, based on individual preferences

At the end of the 10 sessions, which lasted about five weeks, a re-elaboration of the treatment was carried out, together with the participant, in which the emotions, experiences, perceptions, and gratification felt in participating in this study were gathered.

### 2.4. Data Analysis

Participants were evaluated at the beginning and the end of the treatment. A descriptive analysis was carried out to obtain general information such as gender, age, comorbidity, years of diagnosis, MMSE score and Hoehn and Yahr score. For inferential analysis, the Wilcoxon signed-rank test was used to calculate the statistical significance of the values obtained from the evaluation scales and tools used. The significance level was set for a *p*-value less than or equal to 0.05. Our 5 hypotheses were tested using Bonferroni adjusted alpha levels of 0.01 per test (0.05/5). When using the Self-rating Depression Scale, Unified Parkinson’s Scale Disease Rating, Rivermead Adl Scale, Scala di Tinetti, and Parkinson’s Disease Questionnaire-3, equine therapy intervention should improve the mental and mood state, the activities of daily life, and motor activity in people with PD. All the statistical analyses were performed by using IBM SPSS Statistics for Windows (Version 23.0; IBM Corp., Armonk, NY, USA).

## 3. Results

In the study, 17 participants who met the inclusion criteria were recruited. All were informed about the study aims and procedures, and all agreed to participate. All the included individuals completed the 10 ET program sessions. General participants’ data were collected, and mean, median and standard deviation and frequency were calculated. The sample was composed of 11 women (64.7%), the mean age of participants was 67.94 ± 10.77, 76.5% of participants were in H&Y stage III, and adherence to treatment was 100%. The descriptive analysis data are shown in Table 1.

The inferential analysis of the results of the administered evaluation tests reported the significant values, which were preceded by the calculation of the median and standard deviation at the beginning (T0) and the end (T1) after five weeks of treatment. Results are reported in Table 2.

The analysis of the data of the stabilometric platform revealed the significant values, which were preceded by the calculation of the median and standard deviation at the beginning and the end of treatment. Measurements were repeated with eyes open and with eyes closed; results are reported in Table 3.

## 4. Discussion

In the last decades, rehabilitative interventions that provide assistance to animals have spread with programs aimed at various diseases and populations. Scientific evidence for ET has reported benefits in children with ASD, infants with infantile cerebral paralysis, and people with multiple sclerosis [9,10,11,40].

The objective of the study was to evaluate the change in occupational performance, quality of life, mood, balance, pace, and ability to perform activities of daily life due to ET in people with PD. From the statistical analysis of the data, it emerged that those who participated in ET reported statistically significant improvements in mood, according to the analysis of the Zung Self-Rating Depression Scale and ADL data, the Rivermead ADL scale and the second sub-scale of the UPDRS. There were statistically significant improvements from the processing of PDQ-39 data, particularly in the categories of mobility, ADL, emotional well-being, stigma and cognitive abilities. A non-statistically significant clinical improvement was shown by items that concerned social support, communication, and physical discomfort. Statistical analysis of the Tinetti balance assessment tool was statistically significant. Bonferroni adjusted alpha levels confirms the effectiveness of ET in the mental and mood state, the activities of daily life and motor activity in people with PD.

In analyzing the two distinct sub-scales, it was seen that both gait and balance reported a statistically significant *p*-value. The improvement of balance did not result from postural assessment data obtained from stabilometric platform, which did not show statistically significant values. Scientific evidence shows that individuals with autism showed improvements in adaptive behaviors and in participation in self-care, low-demand leisure, and social interactions [40]. Recent studies outline that the therapeutic effects of AAT in the improvement of quality of life can be achieved with any human-animal interaction [41,42]. In the literature, individuals with multiple sclerosis benefit from ET in the areas of balance, fatigue, spasticity, and quality of life [9].

The results of our study can be completed with the results of a recent feasibility study that shows in non-hospitalized persons with PD an overall improvement in motor skills, as well as positive changes in spatiotemporal gait variables and ameliorated attention functioning and a decrease in apathy levels were also observed, as well as an increase in perceived quality of life [12] This study adds important results to the existing literature on improving the ADL, quality of life and balance in people with PD who engage in ET.

### Limitations of the Study

For future studies, it is recommended that the sample size be increased and to use a stabilometric platform that is more sensitive to the clinical changes of individuals who undergo treatment as an evaluation tool. It is also advisable to investigate long-term benefits by repeating the evaluations everyone to three months. In future studies, more objective evidence such as brain imaging or neuroprotective factor release in horse-riding practice should be included [43,44,45]. Moreover, correlation with medication should be considered. Finally, in future studies, an extra control, such as a horse-riding simulator group is needed in order to distinguish the therapeutic effects is from the emotional exchange between human-animal interaction or the pure low-limb activity training.

## 5. Conclusions

In this study, it emerged that ET, as a supportive treatment for traditional treatments, is effective for the recovery of occupational performance, activities of daily life, mood, balance, gait, and quality of life in individuals with PD. This paper could be valuable to practitioners who use ET.

## Figures and Tables

**Figure 1 healthcare-10-00561-f001:**
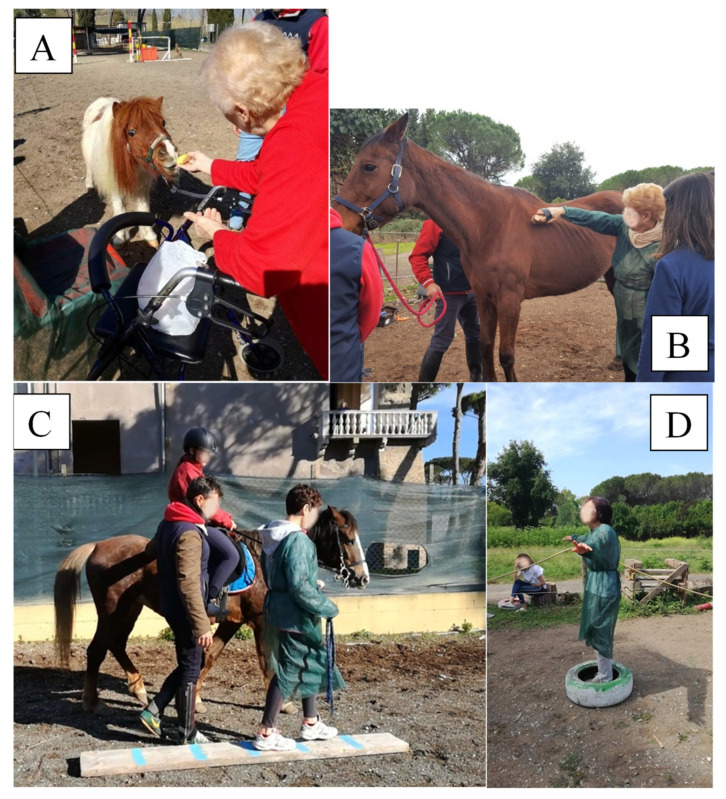
Vocalization exercises (**A**), care and cleaning activity (**B**), horse-assisted walking activity (**C**), and exercises of breathing (**D**) performed by individuals with Parkinson’s disease on the equine therapy program.

**Table 1 healthcare-10-00561-t001:** Demographic characteristics of the 17 individuals with Parkinson’s disease participating to the Equine therapy program.

	Sample 17
Gender male *n*° (%)	6 (35.3%)
Age mean ± SD (range)	67.94 ± 10.77 (52–78)
Years of diagnosis	7.35 ± 3.06
Height mean ± SD (range)	168.13 ± 10.42 (154–188)
Body weight mean ± SD (range)	67.63 ± 13.48 (45–90)
Body mass index mean ± SD (range)	23.84 ± 2.66 (29.40–18.73)
**Comorbidity** *n*° (%)	
Cognitive decline	1 (5.9%)
Pisa syndrome	13 (76.5%)
Cognitive decline and Pisa syndrome	2 (11.8%)
No comorbidity	1 (5.9%)
MMSE mean ± SD	29.88 ± 2.78
**Hoehn and Yahr** *n*° (%)	
Stage 2	4 (23.5%)
Stage 3	13 (76.5%)

**Table 2 healthcare-10-00561-t002:** Median at the beginning and after 10 sessions of equine therapy and calculation of the value of *p* of the administered assessment scales (Self-rating Depression Scale, Unified Parkinson’s Scale Disease Rating, Rivermead Adl Scale, Scala di Tinetti, and Parkinson’s Disease Questionnaire-39).

Rating Scale	T0 Median ± Standard Deviation	T1 Median ± Standard Deviation	Wilcoxon Signed-Rank Test	Statistical Significance	Effect Size	Power Analysis
SDS	42.00 ± 7.45	34.00 ± 6.84	−3.212	*p* < 0.001 *	−0.55	0.86
UPDRS	16.00 ± 8.45	11.00 ± 7.43	−3.189	*p* < 0.001 *	−0.54	0.85
RAS	30.00 ± 6.12	38.00 ± 4.05	−3.520	*p* < 0.0004 *	−0.60	0.91
Tinetti Total	16.00 ± 4.70	20.00 ± 4.82	−2.911	*p* < 0.004 *	−0.49	0.72
Tinetti Gait	6.00 ± 2.52	8.00 ± 2.37	−2.822	*p* < 0.005 *	−0.48	0.75
Tinetti Balance	9.00 ± 2.91	12.00 ± 2.65	−2.064	*p* < 0.039	−0.35	0.49
PDQ−39 Total	70.00 ± 29.09	52.00 ± 19.77	−3.55	*p* < 0.0003 *	−0.60	0.91
Mobility Items	21.00 ± 11.64	10.00 ± 7.24	−3.578	*p* < 0.0003 *	−0.61	0.92
Daily Life Activities Items	9.00 ± 6.89	6.00 ± 5.16	−3.066	*p* < 0.002 *	−0.52	0.81
Emotional Well-being Items	11.00 ± 5.48	8.00 ± 4.87	−2.778	*p* < 0.005 *	−0.47	0.74
Stigma Items	4.00 ± 3.78	3.00 ± 2.87	−2.023	*p* > 0.043	−0.34	0.47
Social Support Items	8.00 ± 2.45	8.00 ± 2.48	−1.633	*p* > 0.102	−0.28	0.34
Cognitive Skills Items	5.00 ± 3.06	4.00 ± 2.02	−2.483	*p* < 0.013	−0.42	0.64
Communication Items	3.00 ± 2.89	2.00 ± 2.36	−1.913	*p* > 0.056	−0.32	0.43
Physical Discomforts Items	5.00 ± 2.91	6.00 ± 3.43	−1.44	*p* > 0.150	−0.24	0.26

***** *p* < 0.01 Bonferroni adjusted alpha levels.

**Table 3 healthcare-10-00561-t003:** Median at the beginning and after 10 sessions of equine therapy and calculation of the value of *p* for the stabilometric platform with open and closed eyes.

Values			Open Eyes						Closed Eyes			
	T0 Median ± Standard Deviation	T1 Median ± Standard Deviation	Wilcoxon Signed-Rank Test	Statistical Significance	Effect Size	Power Analysis	T0 Median ± Standard Deviation	T1 Median ± Standard Deviation	Wilcoxon Signed-Rank Test	Statistical Significance	Effect Size	Power Analysis
Number of readings taken	3000 ± 0.00	3000 ± 0.00	0	*p* = 1.00	0	0.05	3000 ± 0.00	3000± 0.00	0	*p* = 1.00	0	0.05
Middle coronal center (DX-SX) (unit)	−70 ± 9.85	−4.00 ± 12.83	−0.059	*p* < 0.95	−0.01	0.05	5.00 ± 7.887	2.00 ± 7.64	−0.598	*p* < 0.55	−0.10	0.08
Middle sagittal center (PST-ANT) (unit)	−11.60 ± 17.44	−6.80 ± 21.89	−1.958	*p* < 0.50	−0.33	0.44	−3 ± 18.656	−5.50 ± 12.15	−0.889	*p* < 0.374	−0.15	0.13
Curved length (unit)	560 ± 83.71	600 ± 173.59	−2.073	*p* < 0.038	−0.35	0.49	586 ± 226.18	626 ± 155.14	−1.125	*p* < 0.260	−0.19	0.18
90% confidence ellipse area (unit)	32.00 ± 68.07	35 ± 63.31	−0.593	*p* < 0.553	−0.10	0.08	42 ± 72.729	50 ± 114.55	−1.007	*p* < 0.314	−0.17	0.15
Average speed (unit/ms)	0.00 ± 0.00	0.00 ± 0.00	0	*p* = 1.00	0	0.05	0.00 ± 0.00	0.00 ± 0.00	0	*p* = 1.00	0	0.05
Sampling frequency (Hz)	100 ± 0.00	100 ± 0.00	0	*p* = 1.00	0	0.05	100± 0.00	100 ± 0.00	0	*p* = 1.00	0	0.05
Fundamental harmonic frequency (DX-SX) (Hz)	0.00 ± 0.00	0.00 ± 0.00	0	*p* = 1.00	0	0.05	0.00 ± 0.00	0.00 ± 0.33	−1	*p* < 0.317	−0.17	0.15
Fundamental harmonic frequency (PST-ANT) (Hz)	0.00 ± 0.33	0.00 ± 0.00	−1	*p* < 0.317	−0.17	0.15	0.00 ± 0.00	0.00 ± 0.00	0	*p* < 1.00	0	0.05

## Data Availability

The datasets generated and/or analyzed during the current study are available from the corresponding author on reasonable request.

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
