# Peer review of "The Effectiveness of Equine Therapy Intervention on Activities of Daily Living, Quality of Life, Mood, Balance and Gait in Individuals with Parkinson’s Disease"

_healthcare, 2022, doi:10.3390/healthcare10030561_

Round 1
Reviewer 1 Report
healthcare-1563295
The requirements that I made to the article have been solved.
Author Response
Dear Reviewer, the authors thank you for the comments received
Reviewer 2 Report
In line 225, the author deleted the mistakenly inserted reviewer's comments. However, they forget to delete the [ mark in the citation part. :horse-riding practice should be included[43]-45]" should be corrected to [43-45].
Page 5 is blank? Editorial error?
Other than these, we believe the paper is ready to accept.
Author Response
Dear Reviewer, the authors thank you for the comments received, correction has been made to the text.
Best regards
This manuscript is a resubmission of an earlier submission. The following is a list of the peer review reports and author responses from that submission.
Round 1
Reviewer 1 Report
Does the study have approval from any ethics committee and registration as a clinical trial?
Certain sociodemographic data do not appear, such as the age range of the patients (minimum), the percentage of gender, height and weight, variables that can influence the results.
Was there no exclusion criteria?
How was adherence to treatment?
A control group of patients is missing, where these variables were measured without having performed the intervention.
In the assessment tools section, it is necessary to include the city and state in parentheses, apart from the brand (global postural system).
Nothing is said about the medication that the participants were taking, which clearly influences the measured variables.
In the bibliography, in some references the pages of the article appear within the journal and in others only the year appears.
Author Response
Dear reviewer, thank you for your carefull review. Attached here you can kindly find responses to your comments.
Hoping that the new version is suitable for publication.
Best regards

Reviewer 2 Report
The authors investigate the efficacy of Equine Therapy (ET) on the improvement of daily living activity, quality of life, mood, balance, and the gain in patients with Parkinson's disease (PD). Up to now, there are still no effective drugs that could stop or even reverse the progression of the disease. As an important candidate of alternative intervention for treatment to PD, the effectiveness of ET is significant to the future low-cost and sustainable therapeutic options development. However, some questions should be addressed before publication.
1: The author should include more objective evidence such as brain imaging or neuroprotective factor release in horse-riding practice or human-animal interaction activities in the part of the introduction and discussion. The mechanism of human-animal therapeutic in neurodegenerative disease is complicated. However, adding more objective evidence is better for readers to understand the specificity of ET in PD treatment. People studied the neuronal changes after receiving the ET for PTSD patients (1). Also, some evidence suggested that aerobic training could exert neuroprotective effects on dopaminergic neurons (2-3).
2: How could the author distinguish the therapeutic effects is from the emotional exchange between human-animal interaction or the pure low-limb activity training? Is there any extra control, such as a horse-riding simulator group is needed?
3: Is there any evidence suggesting that the therapeutic effects are specific to the equine or the gross motor function and daily life activity improvement can be achieved with any human-animal interaction?
4: Whether all participants choose to participate because of their personal preference for the human-equine interaction, or, in other words, whether the therapeutic effects are specific to people who have a personal preference for equines?
5: As to selecting output measurements, why did the author not pick up the most updated MDS-UPDRS that included more non-motor function parameters?
(1) Zhu, Xi, et al. "Neural changes following equine‐assisted therapy for posttraumatic stress disorder: A longitudinal multimodal imaging study." Human brain mapping 42.6 (2021): 1930-1939.
(2) Ridgel, Angela L., et al. "Acute effects of passive leg cycling on upper extremity tremor and bradykinesia in Parkinson's disease." The Physician and sportsmedicine 39.3 (2011): 83-93.
(3) Frazzitta, Giuseppe, et al. "Effectiveness of intensive inpatient rehabilitation treatment on disease progression in parkinsonian patients: a randomized controlled trial with 1-year follow-up." Neurorehabilitation and neural repair 26.2 (2012): 144-150.
Author Response
Dear Reviewer, thank you for your careful review. Attached here you can kindly find responses to your comments. Hoping this version is suitable for publicatio.
Best regards

Round 2
Reviewer 2 Report
The author seems mistakenly placed my comments into the updated manuscript, lines 216-221. Please take it away or rephrase it.
Author Response
Dear reviewer, thank you for your comment. Lines 216-221 have been deleted.
Best regards